# LiveSearchBench: An Automatically Constructed Benchmark for Retrieval and Reasoning over Dynamic Knowledge

## Abstract

Evaluating large language models (LLMs) on question answering often relies on static benchmarks that reward memorization and understate the role of retrieval, failing to capture the dynamic nature of world knowledge. We present LiveSearchBench, an automated pipeline for constructing retrieval-dependent benchmarks from recent knowledge updates. Our method computes deltas between successive Wikidata snapshots, filters candidate triples for quality, and synthesizes natural-language questions at three levels of reasoning difficulty, each guaranteed to admit a unique, verifiable answer through SPARQL validation. The pipeline is fully automated, scalable across time, and minimizes human intervention, enabling continual regeneration of temporally grounded benchmarks. Experiments show a pronounced performance drop when models confront facts that post-date pretraining, with the gap most salient on multi-hop queries. Retrieval-augmented methods and larger, instruction-tuned models provide partial gains but fail to close this recency gap. By design, LiveSearchBench shifts evaluation from static memorization toward tasks that require up-to-date retrieval and reasoning, offering a foundation for systematic, long-term assessment of LLMs under evolving knowledge. Data and code are available at LiveSearchBench.

## 1 Introduction

Large language models (LLMs) have demonstrated remarkable progress across diverse natural language processing tasks, with solid performance on prominent search question answering (QA) benchmarks such as Natural Questions (Kwiatkowski et al., 2019), TriviaQA (Joshi et al., 2017), and HotpotQA (Yang et al., 2018). Recent reinforcement learning (RL) methods have further improved headline performance, strengthening the perception that LLMs possess sophisticated reasoning and knowledge-intensive inference capabilities (Jin et al., 2025b; Fan et al., 2025). However, a fundamental limitation persists: most search-oriented benchmarks are static and outdated. Many were collected years ago, raising the risk that answers are encoded in models' parametric memory due to pre-training contamination rather than discovered via retrieval (Wu et al., 2025).

World knowledge is inherently dynamic—news breaks, software versions change, policies evolve, and social events unfold—yet prevailing benchmarks lack mechanisms to incorporate real-time updates. Because of this static nature, evaluating retrieval on these datasets is unreliable: models can often answer questions without invoking any search, relying solely on internal memory. As emphasized by the notion of a *Knowledge Boundary* (Wang et al., 2025; Chen et al., 2025b), there is a critical distinction between what a model remembers and what it must acquire externally. Our preliminary experiments corroborate this concern: several models achieve strong scores even when retrieval is disabled, suggesting that memorized knowledge dominates and obscures true capacity for acquiring and reasoning over up-to-date external information.

To contextualize the evolution of QA evaluation and retrieval-centric resources, Figure 1 highlights key datasets and model milestones. As the timeline shows, many widely used benchmarks predate recent advances in search-integrated inference, and community efforts have largely prioritized model development over evaluation under dynamic conditions. Motivated by these gaps—and inspired by the contamination-aware practices of LiveCodeBench (Jain et al., 2024)—we introduce

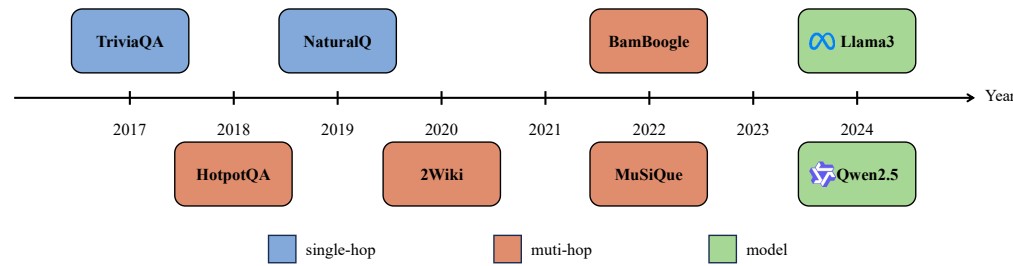

Figure 1: A timeline of major QA benchmarks and model releases. The figure illustrates the historical reliance on static benchmarks, motivating the need for dynamic evaluation resources.

**LiveSearchBench**, a continually updated benchmark built via a scalable pipeline that synthesizes questions from real-world editing streams. The benchmark remains fresh and temporally grounded, with validation that enforces both factual correctness and temporal consistency. By design, success hinges on up-to-date retrieval rather than parametric recall, moving beyond memory-based performance on static snapshots. Our evaluation yields three key insights. First, by systematically testing LLMs and retrieval-augmented generation (RAG) systems on LiveSearchBench, we expose marked differences in their ability to handle dynamic knowledge. Second, we observe a persistent gap between memorization-driven responses and genuine retrieval-based inference. Third, these findings underscore the need for benchmarks that reflect realistic, time-sensitive conditions. This paper makes the following contributions:

◇ We develop a scalable data-generation pipeline that continuously harvests questions from real-world editing streams, coupled with validation that enforces factuality and temporal correctness.

◇ We conduct an extensive evaluation of state-of-the-art LLMs and RAG methods on LiveSearch-Bench, revealing strengths and limitations in handling dynamic, time-sensitive knowledge.

◇ We will release LiveSearchBench as a continually updating resource, enabling the community to track progress on retrieval-augmented methods under realistic, temporally grounded conditions.

## 2 RELATED WORK

**Large Language Models and Search Retrieval.** LLMs leverage external knowledge along three complementary axes (Zhang et al., 2025). (i) Retrieval-augmented generation (RAG). RAG has become a prevailing strategy for grounding outputs in external evidence, and recent surveys consolidate design choices and best practices Gao et al. (2024); Fan et al. (2024). (ii) Workflow-style search agents. Agentic systems explicitly plan queries, browse sources, verify snippets, and synthesize answers; recent efforts integrate these steps into inference-time reasoning traces, exemplified by Search-o1 (Li et al., 2025c). (iii) Reinforcement learning for search and reasoning. RL improves query formulation and the coordination between search and reasoning by directly optimizing end-to-end behavior on challenging objectives (Jin et al., 2025b; Fan et al., 2025; Sun et al., 2025; Song et al., 2025; Chen et al., 2025a). These lines differ in where the search policy resides and how evidence is injected, yielding complementary avenues for strengthening LLMs' use of external knowledge.

**Search QA Benchmarks.** Single-hop search QA is widely evaluated using Natural Questions, TriviaQA, and SimpleQA (Kwiatkowski et al., 2019; Joshi et al., 2017; Wei et al., 2024). Multi-hop reasoning is assessed by HotpotQA, 2WikiMultihopQA, MuSiQue, Bamboogle, and BrowseComp (Yang et al., 2018; Ho et al., 2020; Trivedi et al., 2022; Press et al., 2022; Wei et al., 2025). While SimpleQA and BrowseComp incorporate careful curation and adversarial design, all these resources remain static snapshots. This limits scalability, risks overlap with pre-training corpora, and provides weak coverage of time-sensitive knowledge. In parallel, recent work constructs synthetic, web-grounded data for training (Tao et al., 2025; Li et al., 2025a), —aimed at scaling instruction-tuning or corpus quality rather than evaluation; such pipelines generally do not enforce temporal recency, uniqueness guarantees, or machine-verifiable provenance.

# 3 PRELIMINARY ANALYSIS: INTERNAL MEMORY VS. TOOL-AUGMENTED RETRIEVAL

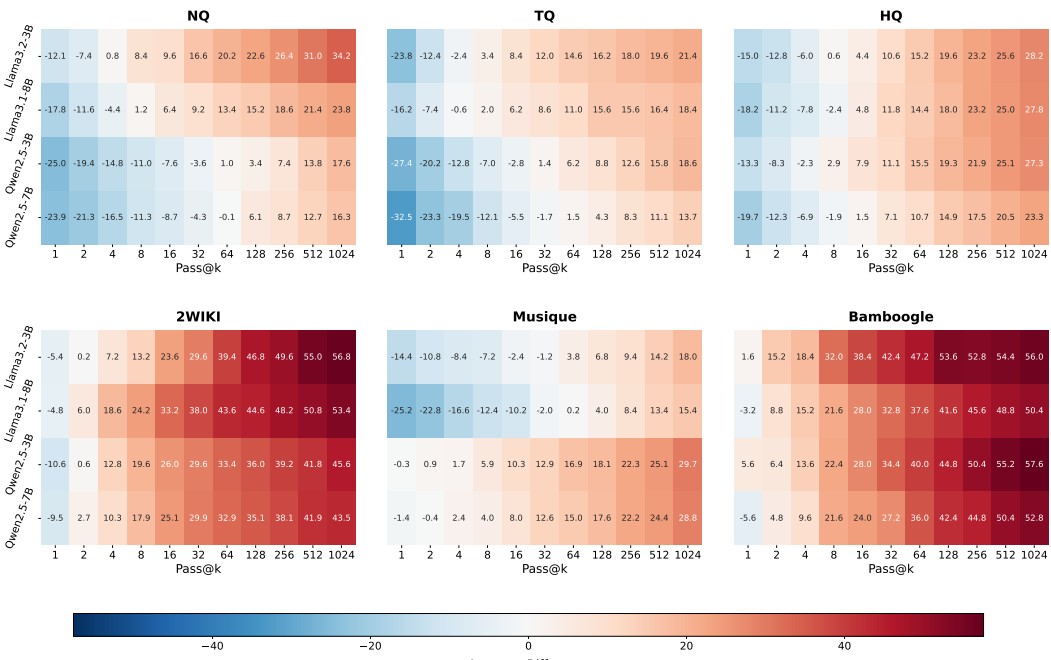

Figure 2: Accuracy difference $\Delta_k = \text{Pass@}k_{\text{no-search}} - \text{Pass@}1_{\text{search}}$ across six QA benchmarks and multiple model sizes. Red regions denote that parametric-only inference outperforms retrieval@1, while blue regions indicate the opposite.

**Benchmark-Level Patterns.** Figure 2 shows systematic differences across datasets. On single-hop benchmarks such as NQ and TQ, retrieval@1 provides limited improvement when $k$ is small, and parametric-only inference rapidly catches up as $k$ increases, suggesting that many answers are already stored in models' internal memory. On multi-hop benchmarks including HotpotQA, 2Wiki, MuSiQue, and Bamboogle, red regions dominate at larger $k$, indicating that retrieval can sometimes introduce distractors or stale evidence, while parametric inference continues to benefit from sampling. Overall, these patterns suggest that static datasets may overstate the role of retrieval tools and understate the extent to which success comes from memorized knowledge.

**Scaling with Sampling.** A direct comparison between retrieval@1 and parametric-only inference (Pass@$N$) reveals that as $N$ grows, sampling without retrieval often matches or surpasses retrieval-based results. This effect is visible in the red-dominated regions of Figure 2. From a benchmark perspective, this highlights that current static QA datasets tend to undervalue retrieval, since models can perform competitively by leveraging stored knowledge combined with sampling strategies.

**Observations.** In the experiment, parametric-only systems can match or even exceed retrieval-augmented pipelines on static datasets—without accessing external evidence. Retrieval is not uni-formly beneficial, especially on multi-hop datasets, where it can introduce noise and compound errors. Additionally, different model families exhibit consistent offsets as $k$ increases. Our experiments show that pass@k accuracy is relatively high even without retrieval, suggesting that with reinforcement learning (RL) techniques, the pass@$k$ score could potentially converge towards pass@1, further closing the gap and possibly surpassing retrieval@1 (Fan et al., 2025; Guo et al., 2025). These patterns validate our hypothesis: static benchmarks overestimate an LLM's ability to handle dynamic, time-sensitive knowledge. They reward distributional familiarity and sampling strategy, rather than the need for up-to-date evidence, underscoring the necessity for benchmarks whose questions explicitly depend on current, verifiable sources.

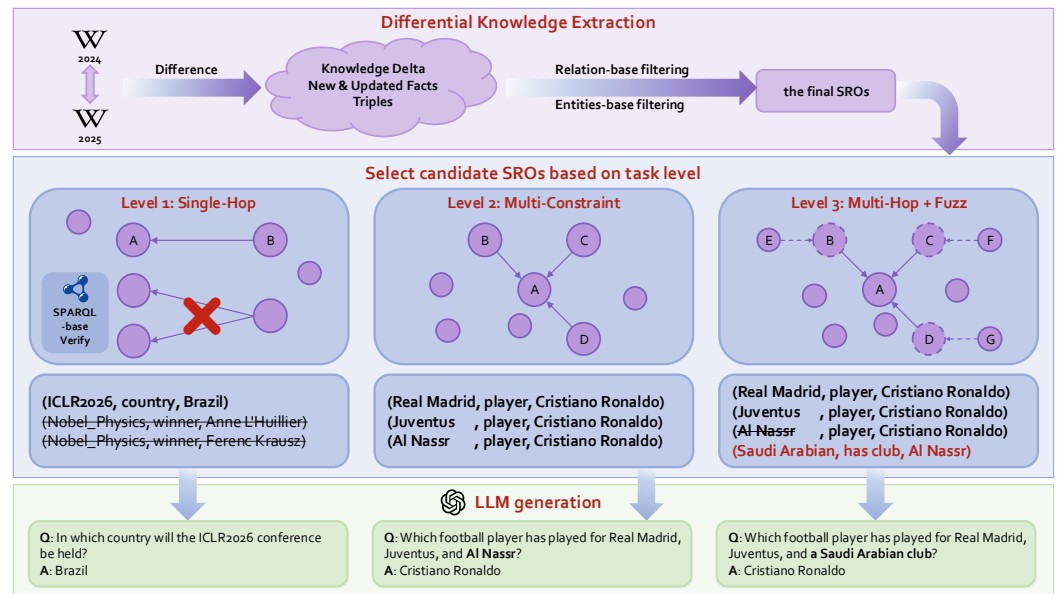

Figure 3: Overview of the generation pipeline. We compute a *knowledge delta* between two Wikidata snapshots to obtain new or updated subject–relation–object (SRO) triples. After relation and entity based filtering, candidate triples are used to synthesize questions at three difficulty tiers: (L1) single-hop, (L2) multi-constraint multi-hop, and (L3) multi-hop with attribute fuzzing. All questions are verified against the current snapshot via SPARQL to ensure correctness.

## 4 LIVESEARCHBENCH

### 4.1 PROBLEM FORMULATION

To address the evolving nature of world knowledge, we propose leveraging the dynamic updates of the Wikidata knowledge graph to construct question-answering (QA) problems. As Wikidata continually incorporates new information, it provides a rich source of facts that can be used to generate up-to-date QA instances. Building on this idea, we formalize QA in the context of dynamic knowledge graphs. Let $\mathcal{G} = (\mathcal{V}, \mathcal{E})$ denote the Wikidata knowledge graph, where $\mathcal{V}$ is the set of entities and literals, and $\mathcal{E}$ is the set of directed triples $(h, r, t)$ with head $h \in \mathcal{V}$, relation $r$, and tail $t \in \mathcal{V}$. A question $q$ is formalized as a constrained path query over $\mathcal{G}$, and the gold answer $a^\star \in \mathcal{V}$ (or a literal) must be *unique* under these constraints.

$$\text{Answer}(q, \mathcal{G}_{T_1}) = a^\star \tag{1}$$

This uniqueness requirement, validated against the snapshot $\mathcal{G}_{T_1}$ via SPARQL queries, ensures that every benchmark instance admits a single, verifiable solution. Consequently, once the *new or updated* triples between two snapshots are extracted, the benchmark can be constructed automatically through a unified pipeline, without the need for manual annotation or domain-specific heuristics.

### 4.2 BENCHMARK DESIGN AND GENERATION PIPELINE

**Design Goals.** Our aim is to build a continually updating benchmark that faithfully reflects the evolving nature of world knowledge. The design is guided by four principles: ① questions should target *recent* facts unlikely to reside in an LLM's parametric memory; ② each instance must admit a *unique*, verifiable answer grounded in a public knowledge base; ③ the benchmark should offer controllable difficulty through structured hop levels; and ④ the pipeline should be *fully automated*, ensuring scalability and sustainability with minimal human intervention. We instantiate these goals on WIKIDATA, leveraging its continually evolving knowledge graph and SPARQL endpoint. This setup guarantees freshness and verifiability while enabling systematic control over reasoning complexity without costly manual curation. Figure 3 presents an overview of our pipeline, which

transforms evolving knowledge in WIKIDATA into retrieval-dependent QA instances. The process is fully automated and proceeds in four main stages. Pseudocode for the full pipeline is provided in Appendix §B.2.

**Step 1: Differential Knowledge Extraction.** We take two Wikidata snapshots at times $T_0$ and $T_1$ ($T_1 > T_0$) and normalize each into a set of SRO triples, $\mathcal{G}_{T_0}$ and $\mathcal{G}_{T_1}$. We then construct the *knowledge delta* as the union of insertions and updates:

$$\Delta^+ = \{\, t \in \mathcal{G}_{T_1} \setminus \mathcal{G}_{T_0} \,\}, \quad \Delta^\circ = \{\, (s,r,o_1) \in \mathcal{G}_{T_0},\ (s,r,o_2) \in \mathcal{G}_{T_1}:\ o_1 \neq o_2 \,\}, \quad \Delta = \Delta^+ \cup \Delta^\circ.$$

Here, $\Delta^+$ captures newly added facts, and $\Delta^\circ$ captures *updated* statements where the object set for a given $(s,r)$ changed between snapshots. Every instance therefore anchors to information that post-dates typical pretraining corpora, discouraging memorization and encouraging retrieval.

**Step 2: Candidate Filtering.** The raw delta may contain noisy or underspecified triples. We apply three filters: (i) Relation allow-list. We exclude non-informative predicates using a curated allow-list. (ii) Entity quality and disambiguation. We require language coverage for labels/aliases, prune entities with incomplete metadata, and remove items whose surface forms are highly ambiguous without additional qualifiers. (iii) Statement validity. We drop deprecated or contradictory statements and deduplicate near-duplicates using normalized keys. The result is a pool of recent, interpretable triples suitable for question synthesis.

**Step 3: Hierarchical Question Synthesis.** From the curated triples, we synthesize questions at three levels, enforcing a single correct answer via SPARQL COUNT=1. L1 (single-hop): directly materialize a triple $(a,r,b)$ and keep it only if $b$ is uniquely identifiable in $\mathcal{G}_{T_1}$. L2 (multi-constraint): start from a target entity and iteratively add attribute constraints (e.g., occupation, country, affiliation), checking after each addition whether uniqueness is achieved; we stop when COUNT=1. L3 (multi-hop with fuzz): extend L2 by (a) relaxing an attribute to a broader type/hypernym ("fuzzing") and (b) appending one relational hop; we verify that, despite fuzzing and the extra hop, the query still resolves to a single answer.

**Step 4: Finalization and Validation.** We render each query into natural language using contemporaneous labels and templates, then perform a final SPARQL verification against the $T_1$ snapshot to re-check uniqueness and temporal validity after rendering and de-duplication. This final check is necessary because alias normalization, template realization, or batch de-duplication can inadvertently alter constraint bindings and reintroduce ambiguity; additionally, late-arriving snapshot updates may occur during long runs. For reproducibility, we log snapshot hashes and timestamps so that every instance is traceable to its underlying state.

Further discussion and examples are provided in Appendix §B.2, where we describe the full pipeline, filtering composition, and synthesis rules in detail.

## 4.3 QUESTION COMPLEXITY LEVELS

As illustrated in Figure 3, we define three levels of difficulty. The L1–L3 hierarchy defines a controlled progression of difficulty: fact retrieval (L1), compositional reasoning (L2), and ambiguity resolution under fuzziness (L3). By enforcing uniqueness of answers in $\mathcal{G}_{T_1}$, the benchmark remains both rigorous and auditable while reflecting real-world query complexity.

**Level-1 (L1): Single-Hop with Uniqueness.** Given a source entity $a \in \mathcal{V}$ and a relation $r \in \mathcal{R}$, the task is to identify the unique target $b$ such that

$$|\{b : (a,r,b) \in \mathcal{E}\}| = 1. \tag{2}$$

For example, if the knowledge delta introduces the triple (ICLR2026, *country*, Brazil), the corresponding L1 question is: *"In which country will the ICLR2026 conference be held?"* L1 primarily evaluates factual recall of newly introduced triples.

**Level-2 (L2): Multi-Hop via Constrained Intersection.** To model compositional reasoning, we construct queries where two or more relational paths must intersect in exactly one entity:

$$S_1 = \{x \mid (a,r_1,x) \in \mathcal{E}\}, \quad S_2 = \{x \mid (a',r_2,x) \in \mathcal{E}\}, \quad |S_1 \cap S_2| = 1. \tag{3}$$

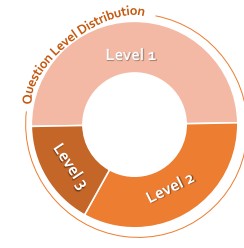
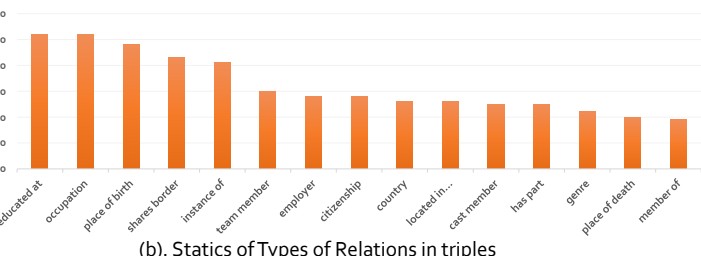

(a).Question Level Distribution       (b). Statics of Types of Relations in triples

Figure 4: Dataset statistics of LIVESEARCHBENCH. (a) Distribution of questions across difficulty tiers L1–L3. (b) Frequency of the most common relation types in synthesized triples. Together, these plots illustrate both the diversity of reasoning requirements and the breadth of relation coverage in our benchmark.

For instance, given the triples (Real Madrid, *player*, Cristiano Ronaldo), (Juventus, *player*, Cristiano Ronaldo), and (Al Nassr, *player*, Cristiano Ronaldo), the benchmark synthesizes the question: *"Which football player has played for Real Madrid, Juventus, and Al Nassr?"* The uniqueness of the intersection ensures that the answer is well defined.

**Level-3 (L3): Attribute Fuzzing with an Additional Hop.**  L3 raises difficulty by deliberately enlarging candidate sets through fuzzing and then adding a disambiguating constraint. Formally,

$$S_1' = \text{fuzz}(S_1), \quad S_2' = \text{fuzz}(S_2), \quad |S_1' \cap S_2' \cap S_3| = 1. \tag{4}$$

For example, consider (Real Madrid, *player*, Cristiano Ronaldo), (Juventus, *player*, Cristiano Ronaldo), and (Al Nassr, *player*, Cristiano Ronaldo). Instead of fixing Al Nassr, we fuzz it into the broader category "a Saudi Arabian club," represented by (Saudi Arabian, *has club*, Al Nassr). The resulting question becomes: *"Which football player has played for Manchester United, Real Madrid, Juventus, and a Saudi Arabian club?"* This fuzzing step broadens the candidate pool, while the added constraint ensures a unique answer.

### 4.4 DATASET COLLECTION

To build the benchmark, we applied our pipeline to two pairs of Wikidata snapshots. For the recent setting, we used the May 2025 and August 2025 dumps to create LIVESEARCHBENCH-2025; for the historical setting, we used the September 2021 and December 2021 dumps to create LIVESEARCHBENCH-2021. In both cases, all instances are grounded in facts that appeared strictly after the earlier snapshot, ensuring temporal recency and reducing overlap with pretraining data. While the pipeline can generate much larger datasets, we opted for a cost-efficient representative subset: 150 L1, 100 L2, and 50 L3 questions. This stratified sample balances reasoning diversity with evaluation efficiency and suffices for robust comparative analysis. Dataset statistics for LIVESEARCHBENCH are shown in Figure 4, illustrating the distribution of questions across difficulty tiers (L1–L3) and the variety of relation types in the synthesized triples. To guarantee quality, five PhD researchers reviewed the synthesized triples and reasoning paths behind each question. Their inspection confirmed the validity and clarity of the 600 questions set, establishing LIVESEARCHBENCH as a reliable evaluation resource.

## 5 EXPERIMENTS

### 5.1 EXPERIMENTAL SETUP

**Datasets.**  We evaluate models on two benchmark instances generated by our pipeline, stratified across the three difficulty tiers (L1, L2, L3). The primary evaluation metric is *Exact Match (EM)* accuracy, requiring a prediction to exactly match the gold answer string. To examine the role of knowledge recency, we construct two batches: 2021 Batch: derived from knowledge updates between September and December 2021. These facts likely overlap with pretraining corpora of many

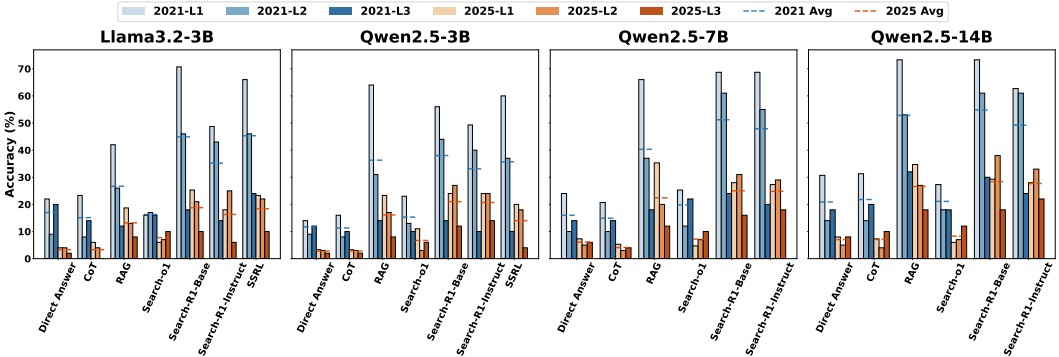

Figure 5: Performance of different models across methods and difficulty levels. Blue bars: 2021 batch; orange bars: 2025 batch;dashed lines: average accuracy.

| Model & Method | LiveSearchBench2021 | | | | LiveSearchBench2025 | | | |
|---|---|---|---|---|---|---|---|---|
| | L1 | L2 | L3 | Avg. | L1 | L2 | L3 | Avg. |
| **Llama3.2-3B-Instruct** | | | | | | | | |
| Direct Answer | 22.0 | 9.0 | 20.0 | 17.0 | 4.0 | 4.0 | 2.0 | 3.3 |
| CoT | 23.3 | 8.0 | 14.0 | 15.1 | 6.0 | 4.0 | 0.0 | 3.3 |
| RAG | 42.0 | 26.0 | 12.0 | 26.7 | 18.7 | 13.0 | 8.0 | 13.2 |
| Search-o1 | 16.0 | 17.0 | 16.0 | 16.3 | 6.0 | 7.0 | 10.0 | 7.7 |
| Search-R1-Base | 70.7 | 46.0 | 18.0 | 44.9 | 25.3 | 21.0 | 10.0 | 18.8 |
| Search-R1-Instruct | 48.7 | 43.0 | 14.0 | 35.2 | 18.0 | 25.0 | 6.0 | 16.3 |
| SSRL | 66.0 | 46.0 | 24.0 | 45.3 | 23.3 | 22.0 | 12.0 | 18.4 |
| **Qwen2.5-3B-Instruct** | | | | | | | | |
| Direct Answer | 14.0 | 9.0 | 12.0 | 11.7 | 3.3 | 3.0 | 2.0 | 2.8 |
| CoT | 16.0 | 8.0 | 10.0 | 11.3 | 3.3 | 3.0 | 2.0 | 2.8 |
| RAG | 64.0 | 31.0 | 14.0 | 36.3 | 23.3 | 17.0 | 8.0 | 16.1 |
| Search-o1 | 23.0 | 13.0 | 10.0 | 15.3 | 11.0 | 3.0 | 6.0 | 6.7 |
| Search-R1-Base | 56.0 | 44.0 | 14.0 | 38.0 | 24.0 | 27.0 | 12.0 | 21.0 |
| Search-R1-Instruct | 49.3 | 40.0 | 10.0 | 33.1 | 24.0 | 24.0 | 14.0 | 20.7 |
| SSRL | 60.0 | 37.0 | 10.0 | 35.7 | 20.0 | 18.0 | 4.0 | 14.0 |
| **Average across models** | 43.9 | 27.6 | 14.9 | 28.8 | 15.6 | 13.5 | 7.4 | 12.2 |

Table 1: **Exact match accuracy (%) on the 2021 and 2025 batches of LiveSearchBench for smaller-scale models.** Results for **Llama3.2-3B** and **Qwen2.5-3B** show that retrieval-augmented methods consistently outperform direct prompting and CoT. Nonetheless, accuracy drops sharply in the 2025 batch, underscoring the challenge of reasoning over genuinely novel knowledge.

baseline models, representing a *seen-knowledge* condition. 2025 Batch: derived from updates between May and August 2025. These facts post-date training cutoffs of current LLMs, representing *novel knowledge* beyond parametric memory.

**Baseline Methods.** We group baselines into three categories. Vanilla Prompt Methods include Direct Prompt and Chain-of-Thought (CoT) prompting to elicit structured reasoning without external evidence. RAG-based Methods comprise standard retrieval-augmented generation and SEARCH-O1 (Li et al., 2025b). RL-based Methods include SEARCH-R1 (Jin et al., 2025a), and SSRL (Fan et al., 2025). To ensure a fair comparison in online settings, the number of retrieved passages is capped at 3 across all RAG-style approaches. For vanilla prompt methods, we employ instruction-tuned variants because they exhibit stronger prompt-following behavior. Full implementation details, hyperparameters and some other baselines are provided in Appendix C and code repo.

| | LiveSearchBench2021 | | | | LiveSearchBench2025 | | | |
|---|---|---|---|---|---|---|---|---|
| **Model & Method** | **L1** | **L2** | **L3** | **Avg.** | **L1** | **L2** | **L3** | **Avg.** |
| **Qwen2.5-7B-Instruct** | | | | | | | | |
| Direct Answer | 24.0 | 10.0 | 14.0 | 16.0 | 7.3 | 5.0 | 6.0 | 6.1 |
| CoT | 20.7 | 10.0 | 14.0 | 14.9 | 5.3 | 3.0 | 4.0 | 4.1 |
| RAG (Standard) | 66.0 | 37.0 | 18.0 | 40.3 | 35.3 | 20.0 | 12.0 | 22.4 |
| Search-o1 | 25.3 | 12.0 | 22.0 | 19.8 | 4.7 | 7.0 | 10.0 | 7.2 |
| Search-R1-Base | 68.7 | 61.0 | 24.0 | 51.2 | 28.0 | 31.0 | 16.0 | 25.0 |
| Search-R1-Instruct | 68.7 | 55.0 | 20.0 | 47.9 | 27.3 | 29.0 | 18.0 | 24.8 |
| **Qwen2.5-14B-Instruct** | | | | | | | | |
| Direct Answer | 30.7 | 14.0 | 18.0 | 20.9 | 8.0 | 5.0 | 8.0 | 7.0 |
| CoT | 31.3 | 14.0 | 20.0 | 21.8 | 7.3 | 4.0 | 10.0 | 7.1 |
| RAG (Standard) | 73.3 | 53.0 | 32.0 | 52.8 | 34.7 | 27.0 | 18.0 | 26.6 |
| Search-o1 | 27.3 | 18.0 | 18.0 | 21.1 | 6.0 | 7.0 | 12.0 | 8.3 |
| Search-R1-Base | 73.3 | 61.0 | 30.0 | 54.8 | 29.3 | 38.0 | 18.0 | 28.4 |
| Search-R1-Instruct | 62.7 | 61.0 | 24.0 | 49.2 | 28.0 | 33.0 | 22.0 | 27.7 |
| **Average across models** | 49.5 | 34.6 | 21.3 | 35.1 | 18.8 | 21.0 | 11.0 | 16.9 |

Table 2: **Exact match accuracy (%) on the 2021 and 2025 batches of LiveSearchBench for larger-scale Qwen models.** Compared to the 3B counterparts in Table 1, both **Qwen2.5-7B** and **Qwen2.5-14B** achieve stronger performance across all difficulty levels, particularly under retrieval-augmented settings. However, performance degradation in the 2025 batch remains evident, high-lighting that scale alone cannot fully compensate for the challenge of unseen, dynamic knowledge.

## 5.2 MAIN RESULTS

We assess performance across the two temporal batches (2021, 2025), the three levels (L1–L3). Our analysis centers on four themes: (i) recency effects, (ii) the benefit of retrieval, (iii) family/scale effects, and (iv) level-wise trends. We visualize the main results in Figure 5, Table 1 and Table 2.

**Retrieval vs. No Retrieval.** To assess the role of retrieval, we compare average exact-match accuracy between vanilla prompting methods (Direct Answer, CoT) and retrieval-based methods (RAG, Search-o1, Search-R1, SSRL). Figure 6 visualizes this difference via absolute improvement and relative gain, confirming that dynamic evaluation more clearly exposes the necessity of retrieval tools. On the 2021 batch, retrieval yields only moderate improvements, consistent with many facts already being encoded in model parameters. In contrast, the 2025 batch shows a substantially larger advantage, demonstrating that retrieval is indispensable when addressing genuinely new knowledge absent from pretraining corpora. Beyond absolute accuracy gains, retrieval also delivers much higher relative improvements in 2025, underscoring models' growing reliance on external evidence.

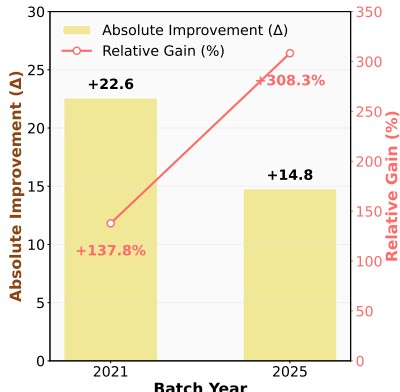

Figure 6: Absolute (Δ) and relative (%) improvements of retrieval-based methods over Direct Answer, averaged across models, on the 2021 vs. 2025 batches.

**Batch Comparison:** Across all models, performance on the **2021 batch** is consistently higher than on the **2025 batch**. Since both datasets are constructed through the same automated pipeline, part of this gap may stem from incidental difficulty differences in the sampled questions. Nevertheless, the magnitude of degradation suggests that *novel knowledge*—facts emerging after

pretraining—poses a substantially greater challenge for LLMs. This highlights the importance of evaluating models under temporally dynamic settings, where internal memorization is insufficient.

**Model Comparison** A cross-family comparison reveals a clear shift, shown in Figure 7. At the scale of 3B and the 2021 batch, Llama3.2-3B consistently outperforms Qwen2.5-3B across nearly all tiers, likely reflecting stronger alignment with older knowledge. However, this advantage diminishes on the 2025 batch: Qwen models, especially under retrieval-based methods, often match or surpass Llama, suggesting stronger adaptation to emerging facts through evidence integration. This contrast highlights two dynamics: (i) pretraining overlap favors Llama on older data, while (ii) retrieval robustness benefits Qwen on newer data. Together, these trends underscore how model families differ not only in baseline knowledge coverage but also in their ability to leverage retrieval for generalization. As the size of the Qwen model family grows, its performance on the dataset continues to improve.

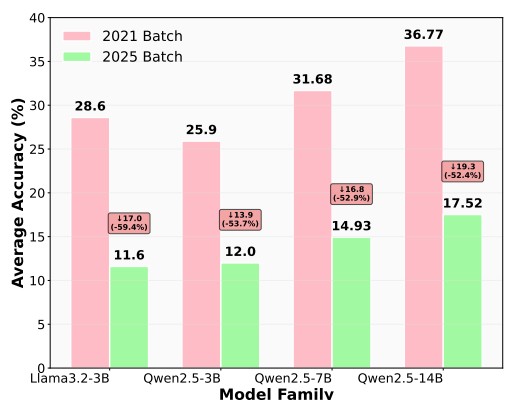

Figure 7: **Family-level comparison.** Averages for different models on 2021 and 2025 batches.

**Effect of Model Scale.** Scaling up from 3B to 7B/14B yields consistent gains in both static (2021) and dynamic (2025) settings. Larger models are particularly more capable in retrieval-augmented configurations, where they can better integrate external evidence. Nonetheless, the gap between the two batches persists even at 14B, showing that model size alone cannot overcome the limitations imposed by knowledge recency.

## 5.3 ANALYSIS

**Trends Across Difficulty Levels.** Across both 2021 and 2025, accuracy typically declines from L1 to L3, reflecting the greater sensitivity of multi-constraint and multi-hop queries to stale passages and distractor evidence. In 2025 we also observe cases where **L1 averages fall below L2**. We attribute this to a rare-entity effect: L1 is seeded by single triples with minimal constraints and thus disproportionately targets rare or newly introduced entities with sparse coverage in external indexes, whereas L2's additional attributes help focus retrieval on the correct target without altering the underlying answer. Crucially, every instance in our benchmark is verified to have a unique answer via SPARQL (`COUNT=1`) against the snapshot $\mathcal{G}_{T_1}$, so this phenomenon is not due to question ambiguity but rather to differences in retrieval precision under rarity and recency. These observations suggest that evaluation should calibrate by entity frequency in addition to hop count and constraint depth, and that retrieval pipelines may benefit from freshness-aware indexing and alias/qualifier normalization when handling rare, recent entities.

## 6 CONCLUSION

We introduced LIVESEARCHBENCH, a continually updated benchmark for evaluating large language models under dynamic knowledge conditions. Experiments reveal a pronounced performance drop when models confront facts that post-date pretraining, with the gap most salient on multi-hop queries. Retrieval-augmented methods and larger, instruction-tuned models deliver partial gains but do not close the recency gap, highlighting the limits of static, memory-friendly QA evaluation. These findings motivate protocols that explicitly depend on up-to-date evidence and assess the coordination between search and reasoning. We intend LIVESEARCHBENCH to serve as a foundation for methods that couple real-time retrieval with stronger reasoning and continual adaptation to evolving knowledge.

ETHICS STATEMENT

This work leverages *publicly available* Wikidata snapshots as the sole knowledge source. Wikidata is collaboratively maintained under open licenses, and our pipeline only processes structured triples that are already public. No personal, sensitive, or proprietary data are involved, and all derived benchmark questions are grounded in verifiable facts with explicit provenance. Because our method is fully automated and does not require human annotations or crowdsourcing, there are no risks of exploitation or privacy leakage. We emphasize that LIVESEARCHBENCH is intended purely for the evaluation of large language models, not for deployment in real-world decision-making scenarios. To the best of our knowledge, this study raises no ethical concerns regarding human subjects, animal welfare, or data misuse.

REPRODUCIBILITY STATEMENT

We have prioritized reproducibility in both benchmark construction and experimental evaluation. All code implementing the data pipeline, including differential extraction, filtering, question synthesis, and validation, will be released under an open-source license. To ensure transparency, we provide snapshot identifiers, hashes, and timestamps, enabling exact regeneration of benchmark instances from raw Wikidata dumps. The specific datasets used in this paper (LIVESEARCHBENCH-2021 and LIVESEARCHBENCH-2025) will be publicly available, along with scripts for constructing new instances from future snapshots. Full experimental details—including model variants, inference settings, retrieval configurations, and hyperparameters—are documented in the appendix. These measures collectively ensure that independent researchers can reproduce our benchmarks and results, and extend them to new temporal settings with minimal effort.

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

# LiveSearchBench Supplementary Material

## A    USE OF LARGE LANGUAGE MODELS (LLMS)

We used large language model (ChatGPT) as an assistive tool in two ways: (1) for writing assistance, including language editing and improving the clarity of the manuscript, and (2) for technical support during code environment setup and debugging, particularly when resolving environment-related errors. The model was not used for generating research ideas, designing methodologies, conducting experiments, or analyzing results. All outputs from the LLM were manually verified by the authors, and final decisions regarding both the research content and the manuscript were made by the authors. The authors take full responsibility for the entirety of this work.

## B    PIPELINE DETAILS

### B.1    FILTERING PROCEDURE

The filtering procedure consists of three main steps to ensure high-quality and interpretable relations for QA. We maintain a curated filter-list that excludes meta/formatting predicates, focusing on retaining only those relations that yield interpretable QA. A detailed list of the excluded predicates is provided in Table 3. To ensure comprehensive language coverage, we prune entities with incomplete metadata and remove those with highly ambiguous surface forms unless additional qualifiers are available to clarify the context. Additionally, we eliminate deprecated or contradictory statements and deduplicate near-duplicate entries by normalizing keys. The preferred method for normalization is using the statement ID, but if unavailable, we rely on a combination of $(s, r)$ along with label normalization.

### B.2    FINALIZATION AND VALIDATION PSEUDOCODE AND SPARQL TEMPLATES

```
1  SELECT ?b WHERE {
2    wd:Q_a wdt:P_r ?b .
3    # Optional: Apply filters for rank and time validity.
4  } LIMIT 2
```

Listing 1: SPARQL sketch for an L1 query. The instance is accepted only if the query returns exactly one result.

```
1  SELECT ?x WHERE {
2    { wd:Q_a  wdt:P_r1 ?x .  FILTER(phi_1(?x)) }
3    UNION
4    { wd:Q_a' wdt:P_r2 ?x .  FILTER(phi_2(?x)) }
5  } GROUP BY ?x HAVING (COUNT(?x)=2)
```

Listing 2: SPARQL sketch for an L2 query. The HAVING clause ensures that ?x satisfies both constraints.

```
1  SELECT ?x WHERE {
2    { wd:Q_a  wdt:P_r1 ?x .  FILTER(phi_1_fuzzy(?x)) }
3    UNION
4    { wd:Q_a' wdt:P_r2 ?x .  FILTER(phi_2_fuzzy(?x)) }
5    UNION
6    { ?x      wdt:P_r3 C_c . FILTER(phi_3(?x)) }
7  } GROUP BY ?x HAVING (COUNT(?x)>=3)
```

Listing 3: SPARQL sketch for an L3 query with fuzzy constraints and an additional hop.

```python
def generate_benchmark(dump_T0, dump_T1):
    # 1. Differential Knowledge Extraction
    G_T0 = extract_triples(dump_T0)
    G_T1 = extract_triples(dump_T1)
    knowledge_delta = G_T1.difference(G_T0)

    # 2. High-Quality Candidate Filtering
    curated_delta = filter_triples(knowledge_delta,
                                   rules=['relation_type', 'entity_quality'
, 'statement_rank'])

    # 3. Hierarchical Question Synthesis from recent facts
    benchmark = []
    for seed_triple in curated_delta:
        # Attempt to build questions of increasing difficulty
        question = None
        if not question:
            question = synthesize_question(seed_triple, G_T1, level='L1')
        if not question:
            question = synthesize_question(seed_triple, G_T1, level='L2')
        if not question:
            question = synthesize_question(seed_triple, G_T1, level='L3')

        # 4. Finalization
        if question and is_valid(question):
            final_instance = render_and_finalize(question)
            benchmark.append(final_instance)

    return benchmark

def synthesize_question(triple, graph, level):
    # Builds a SPARQL query based on the level and seed triple.
    # For L2/L3, this involves finding additional constraining triples.
    query = build_sparql_query(triple, graph, level)

    # Validates that the query has a unique answer in the new graph.
    if is_unique_in_graph(query, graph):
        return (query, triple.answer)
    return None
```

Listing 4: End-to-end pipeline for generating benchmark questions from snapshots.

Table 3: The curated filter-list of excluded meta/formatting predicates.

| Property ID | Description |
| --- | --- |
| P18 | image |
| P31 | instance of (often too basic) |
| P279 | subclass of |
| P373 | Commons category |
| P443 | pronunciation audio |
| P460 | said to be the same as |
| P856 | official website |
| P910 | topic's main category |
| P973 | described at URL |
| P1151 | topic's main Wikimedia portal |
| P1343 | described by source |
| P1424 | topic's main template |
| P1559 | name in native language |
| P1629 | Wikidata property |
| P1630 | formatter URL |
| P1659 | related property |
| P1687 | Wikidata property |
| P1696 | inverse property |
| P1705 | native label |
| P1793 | regular expression |
| P1855 | Wikidata property example |
| P1889 | different from |
| P1921 | URI template |
| P2302 | property constraint |
| P2700 | protocol |
| P2875 | property for this type |
| P2916 | source website for the property |
| P2959 | permanent duplicated item |
| P3254 | property usage tracking category |
| P3709 | unit symbol |
| P3713 | pronunciation audio |

# C   IMPLEMENTATION DETAILS

## C.1   IMPLEMENTATION OF BASELINES

For ZeroSearch, Search-R1, and SSRL, we set the temperature to 0.7, and the max response length to 4096. We do not restrict their max turns for search, so that they can search as many times as they want. We use Exact Match (EM) as our evaluation metric. The prompt we use is listed in Table 4. We use Google Search via Serper API for all all them to ensure fairness.

Table 4: Prompt template. The question is appended at the end during training and inference.

**Prompt Template**

Answer the given question. You must conduct reasoning inside `<think>` and `</think>` first every time you get new information. After reasoning, if you find you lack some knowledge, you can call a search engine by `<search>` query `</search>`, and you should return the top searched results between `<information>` and `</information>`. You can search as many times as you want. If you find no further external knowledge needed, you can directly provide the answer inside `<answer>` and `</answer>` without detailed illustrations. For example, `<answer>` Beijing `</answer>`. Question:

# D    ADDITIONAL ANALYSIS: MODEL FAMILY COMPARISON (LLAMA VS. QWEN)

A clear divergence emerges between the Llama and Qwen families across six benchmarks. On single-hop datasets (NQ, TQ, HQ), both families benefit from increased sampling, but Llama models enter the positive regime of $\Delta_k$ earlier and with steeper gains; Qwen retains larger blue regions at small/medium $k$, indicating a slower shift from retrieval reliance to parametric dominance. The difference is more pronounced on multi-hop datasets (2Wiki, Bamboogle): Llama shows deep red saturation across most of the $k$ range, while Qwen improves with $k$ but with smaller margins and a less abrupt transition. MuSiQue shows a slower transition overall, yet the pattern holds. Scaling within each family reinforces this trend: larger Llama models show sharper improvements in $\Delta_k$ than their Qwen counterparts, suggesting that static QA benchmarks disproportionately reward Llama's parametric capacity, whereas Qwen requires larger sampling budgets to approach similar performance.

# E    ADDITIONAL IMPLEMENTATION DETAILS

For detailed code and case, please visit our repository: LIVESEARCHBENCH Repository

