# OpenReview forum: "LiveSearchBench: An Automatically Constructed Benchmark for Retrieval and Reasoning over Dynamic Knowledge"
_ICLR.cc/2026/Conference — ICLR 2026 Conference Withdrawn Submission_

### Official Review · Reviewer_wUJ9 · 2025-10-29

**Soundness:** 3
**Presentation:** 3
**Contribution:** 2
**Rating:** 2
**Confidence:** 3

**Summary:**

The paper proposes LiveSearchBench, an automated pipeline for constructing a dataset that provides a clean testbed for evaluating retrieval-augmented generation. The dataset is created by taking the difference between two successive Wikidata snapshots; low-quality tuples are discarded before converting to QA pairs. The resulting dataset includes three difficulty levels. Evaluation is conducted on two model families (Llama and Owen) with two dataset versions (2021 and 2025). Results show that retrieval yields larger gains on fresher knowledge that is unlikely to be stored in model parameters.

**Strengths:**

* The paper addresses an important problem: once a benchmark is published, models may train on it. An ever-evolving dataset with the latest knowledge is crucial for correctly evaluating RAG systems.
* The benchmark creation process is rigorous with solid quality control. Low-quality triples are filtered in the Candidate Filtering step, improving reliability, and the Finalization and Validation step further improves question quality.

**Weaknesses:**

* Missing discussion of closely related work. The idea appears very similar to AntiLeakBench [1], especially in methodology: both use Wikipedia as the knowledge base, leverage edit history across snapshots to construct non-contaminated data, and use fully automated QA generation pipelines. The differences between this paper and AntiLeakBench should be clarified.
* Limited model families and sizes. It would be better to include closed-source models and more open-source families.
* The “fully automated” claim may be overstated. In §4.4 the authors note: “To guarantee quality, five PhD researchers reviewed the synthesized triples and reasoning paths behind each question,” indicating a human evaluation component.


[1] Wu, X., Pan, L., Xie, Y., Zhou, R., Zhao, S., Ma, Y., Du, M., Mao, R., Luu, A. T., & Wang, W. Y. (2025). AntiLeakBench: Preventing Data Contamination by Automatically Constructing Benchmarks with Updated Real-World Knowledge, arXiv:2412.13670.

**Questions:**

Writing note:
* Define SPARQL at first mention; adding a brief description would improve clarity.

---

### Official Review · Reviewer_sRfe · 2025-11-01

**Soundness:** 2
**Presentation:** 3
**Contribution:** 1
**Rating:** 0
**Confidence:** 5

**Summary:**

This paper introduces LiveSearchBench, an automated pipeline for constructing dynamic knowledge benchmarks in the QA format to test LLM on new knowledge. It builds on the difference of Wikipedia graph snapshots at different times, and constructs QA pairs of different difficulties based on number of hops and entity scopes. Experiments on two samples of the benchmarks show LLM memorization on older Wikipedia data and RAG helps with new information.

**Strengths:**

- The paper is clear and easy to read.

- The benchmark construction based on Wikipedia graph updates makes sense to reduce LLM memorization of the knowledge being tested.

- Results are well presented, and agree with expectations.

**Weaknesses:**

- The novelty of the paper is limited. Dynamic benchmarking is an important direction, but there has been many works such as AntiLeakBench [1], Daily Oracle [2], among others [3, 4]. Among them, one of the most relevant work, AntiLeakBench [1], also works with the Wikipedia updates for constructing knowledge testing samples that are dynamic and not memorized. Compared to the prior research, the proposed work lacks both scale and comprehensiveness in both benchmark construction and evaluation. Plus, none of the relevant prior works have been mentioned at all in the paper.

- The discussion of the dynamic QA benchmarks is also lacking. For example, previous dynamic benchmarks such as Realtime QA [5], Streaming QA [6], Fresh QA [7], etc. are not mentioned and compared. All of the prior efforts aim to construct dynamic knowledge benchmarks in the QA format, from different sources of information. The proposed benchmark adds little to what we had considering the previous research, and its effort and contribution is limited for a full ICLR paper publication.

- The paper is fairly clear, and the presentation is good. In terms of quality, it feels the analysis, benchmarking and evaluation are a bit contrived and toy, which makes it hard to believe this benchmark is going to make a good impact. For example, Figure 1 does not seem to be very helpful with the vast majority of more relevant research work (as mentioned above) not discussed, and it does not add to the point that the paper is making. With the fast pace of NLP/LLM research, the discussion is out of context of the most recent research. The analysis in Section 3 also does not add much to the paper - it is well expected that many of the old datasets are memorized by the LLMs, and the direction of constantly building dynamic benchmarks is an important one without this contrived analysis.

- The message sent by the paper is not clear. It talks about dynamic benchmarking based on Wikipedia updates, and also conducts experiments with retrieval-augmented methods. It builds an 2021 benchmark and a 2025 benchmark, where the 2021 benchmark is memorized by the model. So the 2021 benchmark saw a higher accuracy from LLMs, and retrieval helps more on the newer 2025 benchmark since the knowledge is not in the model parameters. These are all expected results, and the main contribution remains unclear in terms of what it brings to the community. It feels like the paper pieces a few different things together without a clear message, especially given the prior efforts in this space, among which similar ideas have been well executed.

- The benchmark is based on defined relation types and exact match for evaluation, limiting its practical usage for language variations and knowledge complexity. The size of the benchmarks is also small, 300 examples for each diff of Wikipedia graph. Significance and scalability are not demonstrated in the paper, again limiting its practical usefulness.

- The discussion of search and retrieval in Related Work may lack clear logic. RAG is listed with search agents, and RL for search and reasoning, which is hard to parse. RAG is a general approach to leverage external knowledge, and the search results are also passed to LLMs typically for generation, so it is still RAG. It may be that specific approaches use different ways to formulate search queries, search different databases, etc. But they are all RAG for knowledge augmentation. I think the positions of search, RAG, search QA, etc. are not well narrated, rendering the paper a bit less rigorous in its focus and discussions.


> [1] (2024 Dec) AntiLeakBench: Preventing Data Contamination by Automatically Constructing Benchmarks with Updated Real-World Knowledge

> [2] (2024 Nov) ​​Are LLMs Prescient? A Continuous Evaluation using Daily News as the Oracle

> [3] (2024 June) Automating Dataset Updates Towards Reliable and Timely Evaluation of Large Language Models

> [4] (2024 June) Automating Dataset Updates Towards Reliable and Timely Evaluation of Large Language Models

> [5] RealTime QA: What's the Answer Right Now?

> [6] StreamingQA: A Benchmark for Adaptation to New Knowledge over Time in Question Answering Models

> [7] FreshLLMs: Refreshing Large Language Models with Search Engine Augmentation

**Questions:**

1. For the question generation based on the SRO triplets, the natural language is produced with pre-defined templates? Was there any LLM involved?

2. How many Wikipedia updates are there typically from month to month, or from day to day? And out of those, how many can we effectively use for constructing the knowledge testing cases?

---

### Official Review · Reviewer_cErv · 2025-11-01

**Soundness:** 3
**Presentation:** 3
**Contribution:** 3
**Rating:** 6
**Confidence:** 3

**Summary:**

The paper introduces LiveSearchBench, an automatically constructed benchmark for evaluating LLMs’ retrieval and reasoning abilities under dynamic knowledge conditions. The core idea is to leverage temporal deltas between consecutive Wikidata snapshots to automatically synthesize QA instances whose answers depend on newly added or updated facts.
The pipeline consists of four stages: 1) differential knowledge extraction, 2) candidate filtering, 3) hierarchical question synthesis, and 4) SPARQL-based validation. These four stages ensure that every question has a unique, verifiable answer. Three difficulty tiers (L1 to L3) correspond to factual recall, compositional reasoning, and multi-hop reasoning with attribute fuzzing.
Experiments on 600 synthesized QA instances (LiveSearchBench-2021 and LiveSearchBench-2025) show that all tested LLMs experience a significant accuracy drop when confronted with post-training knowledge, especially in multi-hop cases. Retrieval-augmented and RL-based models (e.g., Search-R1, SSRL) mitigate but do not close this "recency gap", highlighting the limits of static benchmarks and the need for temporally grounded evaluation.

**Strengths:**

[S1] Important task with clear motivation. The paper directly addresses a specific underexplored task in current LLM evaluation: overreliance on static benchmarks that reward memorization. By curating questions in post-dated Wikidata changes, LiveSearchBench offers a principled way to isolate retrieval and reasoning over evolving knowledge.

[S2] Methodology is sound and fully automated. The four-stage pipeline (delta extraction -> filtering -> synthesis -> SPARQL verification) is conceptually clean and sound. The strict COUNT=1 uniqueness constraint and final SPARQL re-validation ensure factual correctness without human annotation noise.. The hierarchical difficulty design (L1–L3) also makes the benchmark interpretable and extensible.

[S3] Solid empirical results. Results are systematic and revealing: performance drops 50–60% on the 2025 batch, confirming that current models struggle with unseen, temporally novel information. Retrieval methods like RAG and Search-R1 yield clear gains (+20 absolute EM), and the observed relative improvement (>300%) over direct prompting underscores the framework’s discriminative power.

**Weaknesses:**

[W1] Scale and manual review. Despite claiming full automation, the released set contains only 600 questions with some degree of human verification by five reviewers. This is sufficient for a proof-of-concept but small compared to existing QA benchmarks, potentially limiting statistical reliability and diversity of reasoning types.

[W2] Evaluating more models will make the conclusions more convincing. The experiments focus on a small set of model families (Llama3.2, Qwen2.5) and retrieval-based methods. While results are informative, the scope could be extended to more varied models (e.g., closed-source models) and retrieval infrastructures (web vs. local). Moreover, metrics are limited to Exact Match; additional human or process-level evaluation would clarify reasoning quality versus retrieval precision.

**Questions:**

[Q1] How stable is the SPARQL validation under frequent Wikidata schema changes? Are constraints (e.g., `COUNT=1`) consistently preserved across versions? Are there any nondeterminism in results observed (e.g., the results varying across queries)?

[Q2] Is there a plan to include non-Wikidata deltas (e.g., Wikipedia or Common Crawl) to capture natural-language phrasing and reasoning beyond triples?

---

### Official Review · Reviewer_8yJx · 2025-11-02

**Soundness:** 3
**Presentation:** 2
**Contribution:** 2
**Rating:** 2
**Confidence:** 5

**Summary:**

The paper introduces an automated pipeline to generated factual question answering (QA) benchmarks at any time, based on relations extracted from Wikidata updates.  The goal is to evaluate LLMs' ability to answer questions about facts that may have evolved after their training, by using retrieval augmentation, as well as to use reasoning to combine information from multiple sources. The benchmark is generated by extracting the relevant relations from Wikidata and prompting an LLM to generate QA pairs based on the extracted data. The questions are organized into 3 levels, ones involving a single simple lookup, ones combining information from multiple sources, and ones that combine multiple sources and also require some inference (for example, Ronaldo played for Al Nassr ==> Ronaldo played for a Saudian Arabian club).

The motivation for a frequently updated, automatically generated benchmark is clear, and the general approach appears sound. However, I believe this paper is far from ready for publication (see weaknesses listed below) and so am recommending rejection.

**Strengths:**

1. The work addresses a real need for benchmarks that evolve to stay ahead of LLMs' parametric knowledge.

2. The approach of automatic benchmark generation via updated Wikidata snapshots is sensible (but see a minor concern about Wikidata under "weaknesses").

3. The inclusion of the three question levels is helpful, as it makes it possible to separately evaluate search and reasoning abilities.

**Weaknesses:**

1. There is insufficient analysis of the results.  Even with retrieval, the best performance on the simplest questions (level 1) is in the low 70s%.  I would expect the paper to include analysis of whether the errors come from search errors (this could be done by including an "oracle" search that provides a known correct document) or another source.  In the absence of such details, I am wondering how trustworthy the results are.

2. There is insufficient detail provided about the human validation of the generated questions.  It is stated only that "five PhD researchers reviewed the synthesized triples and reasoning paths".  More information is needed (see questions below).

3. It is hard to judge the quality of the generated QA pairs.  In the absence of convincing human evaluation results, I looked for example QA pairs.  No examples are given beyond the three QA pairs in Fig. 3, and the code base at the provided link also does not appear to include QA pairs.  Also, the code base is incomplete, with multiple missing files (as of this writing, the readme, the code for generation of level 2 and 3 questions, and multiple other files).  Note:  I don't consider the code base to be necessary for acceptance of the paper, but I mention it only because I was hoping to be able to extract QA examples from it.

4. The writing is fairly poor.  While the general pipeline and results using the generated benchmarks are clear, there are some unclear or missing details and vague terms, while the rest of the text is quite repetitive.  For example, the terms "Pass@k_{no_search}", "Pass@1_{search}", "retrieval@1", "Pass@N", "Pass@k" are not defined.  Vague terms include "headline performance", "real-world editing streams", "contemporaneous labels and templates", "reasoning paths" -- all terms for which I can imagine some meaning, but they are not quite clear even in context.  Repetitive text examples:  the first two paragraphs appear to say essentially the same thing, the "three key insights" in the last paragraph of the intro also appear essentially the same, and the observations below Fig. 2 appear to be stated multiple times in different paragraphs.

5. Minor: I believe relations in Wikidata tend to be fairly slow-varying, changing no more frequently than every several months (and indeed the experiments in the paper are based on pairs of snapshots 3 months apart).  Therefore, before a newly trained model can be tested, it may take a few months before a new benchmark version will be able to test it.

6. Minor:  The paper does not mention some of the related prior efforts to build frequently updated factual QA benchmarks, such as FreshQA (https://arxiv.org/abs/2310.03214) and RealTimeQA (https://arxiv.org/abs/2207.13332).  StreamingQA would also be worth mentioning (although it was not, to my knowledge, updated after the paper publication).  I don't think any of these invalidate the contribution of the benchmark, since none of them are fully automatic.  But they deserve mention and comparison in the related work section.

7. Minor: The QA generation relies on ChatGPT (the paper just mentions "an LLM", as far as I can tell, but the codebase uses GPT-5), a proprietary model whose details are unknown and can change at any time.

**Questions:**

Regarding the human evaluation:  Who are the researchers who served as human validators (are any of them the authors or their colleagues?)?  How was the validation done?  Did the validation include only the triples and reasoning paths, or also the final questions?  What were the quantitative results of the evaluation?

---

### Note · Authors · 2025-12-03

I have read and agree with the venue's withdrawal policy on behalf of myself and my co-authors.